# SpoVG Is Necessary for Sporulation in *Bacillus anthracis*

**DOI:** 10.3390/microorganisms8040548

**Published:** 2020-04-10

**Authors:** Meng Chen, Yufei Lyu, Erling Feng, Li Zhu, Chao Pan, Dongshu Wang, Xiankai Liu, Hengliang Wang

**Affiliations:** State Key Laboratory of Pathogens and Biosecurity, Beijing Institute of Biotechnology, 20 Dongdajie Street, Fengtai District, Beijng 100071, China; chenmeng_0710@foxmail.com (M.C.); flygogo.cool@163.com (Y.L.); fengel@sohu.com (E.F.); Jewly54@126.com (L.Z.); panchaosunny@163.com (C.P.); wangdongshu@foxmail.com (D.W.)

**Keywords:** *Bacillus anthracis*, *spoVG*, *spoIIB*, sporulation

## Abstract

The *Bacillus anthracis* spore constitutes the infectious form of the bacterium, and sporulation is an important process in the organism’s life cycle. Herein, we show that disruption of SpoVG resulted in defective *B. anthracis* sporulation. Confocal microscopy demonstrated that a Δ*spoVG* mutant could not form an asymmetric septum, the first morphological change observed during sporulation. Moreover, levels of *spoIIE* mRNA were reduced in the *spoVG* mutant, as demonstrated using β-galactosidase activity assays. The effects on sporulation of the Δ*spoVG* mutation differed in *B. anthracis* from those in *B. subtilis* because of the redundant functions of SpoVG and SpoIIB in *B. subtilis.* SpoVG is highly conserved between *B. anthracis* and *B. subtilis*. Conversely, BA4688 (the protein tentatively assigned as SpoIIB in *B. anthracis*) and *B. subtilis* SpoIIB (SpoIIB_Bs_) share only 27.9% sequence identity. On complementation of the *B. anthracis* Δ*spoVG* strain with *spoIIB_Bs_*, the resulting strain pB*spoIIB_Bs_*/Δ*spoVG* could not form resistant spores, but partially completed the prespore engulfment stage. In agreement with this finding, mRNA levels of the prespore engulfment gene *spoIIM* were significantly increased in strain pB*spoIIB_Bs_*/Δ*spoVG* compared with the Δ*spoVG* strain. Transcription of the coat development gene *cotE* was similar in the pB*spoIIB_Bs_*/Δ*spoVG* and Δ*spoVG* strains. Thus, unlike in *B. subtilis*, SpoVG appears to be required for sporulation in *B. anthracis*, which provides further insight into the sporulation mechanisms of this pathogen.

## 1. Introduction

*Bacillus anthracis*, the causative agent of anthrax, mostly exists in nature in the form of spores, which play an important role in anthrax infection. Sporulation is initiated in harsh conditions, such as nutritional deficiency [1]. The process of *B. anthracis* sporulation is still poorly understood, which has hindered progress towards understanding its impacts on physiology and pathology.

*Bacillus subtilis* has been studied in detail as a model organism for sporulation. The regulation of *B. anthracis* sporulation is generally assumed to be broadly like that of *B. subtilis*. Sporulation in *B. subtilis* has been divided into so-called “stages” 0 to VII using electron microscopy [2,3]—vegetative cells (stage 0), axial filamentation (stage I), asymmetric division (stage II), engulfment (stage III), formation of the cortex and coat (stages IV and V), and spore maturation and mother cell lysis (stages VI and VII) [2,4,5,6]. Asymmetric division is the earliest morphological change that distinguishes a sporulating cell from a nonsporulating stationary-phase cell [7,8]. SpoIIE, an integral membrane protein [9], is involved in asymmetric septum formation in *B. subtilis* and plays a crucial role in sporulation in *B. anthracis*. The *spoIIM* and *spoIIQ* genes play important roles in the early and late stages of engulfment, respectively [10,11]. The *cotE* gene is involved in assembly of the outer coat structure in stage V of sporulation [12,13].

Spore formation in the *B. cereus* group (which includes *B. anthracis*) is similar to that in *B. subtilis*; however, there are some differences. For example, coat assembly begins at the mother-cell-proximal pole of the forespore in *B. subtilis*, whereas coat material first appears on the long axis of the forespore in *B. anthracis* and *B. cereus* [13]. Spo0B of *B. anthracis* has autophosphorylation and ATPase activities, while its ortholog in *B. subtilis* acts only as a phosphotransferase [14].

SpoVG is a pleiotropic regulatory factor in sporulation. In *B. subtilis*, SpoVG negatively regulates asymmetric septum formation and positively regulates cortex formation [7,15,16]. Moreover, SpoVG is involved in the production of hemolysin in *B. subtilis* [15,17]. *B. subtilis* SpoIIB plays a role in engulfment [15,18]. Strikingly, *B. subtilis* single mutants of *spoIIB* or *spoVG* (Δ*spoVG* or Δ*spoIIB*) showed only minor effects on sporulation, while the Δ*spoVG*/Δ*spoIIB* double mutant had severe defects in spore formation at the engulfment stage, with little or no thinning of the septal peptidoglycan [7,11]. However, the function of the *spoVG* gene in *B. anthracis* is almost completely unknown.

In this study, we investigated the function of SpoVG in *B. anthracis* vaccine strain A16R (pXO1^+^, pXO2^−^) [19] by constructing a Δ*spoVG* mutant. We found that in the Δ*spoVG* mutant, sporulation was blocked before asymmetric septum formation. Thus, the function of SpoVG in *B. anthracis* differs from that in *B. subtilis*; in the latter organism, the absence of *spoVG* caused no significant changes in sporulation efficiency because of the redundancy of *spoVG* and *spoIIB* [5,7,20].

## 2. Materials and Methods

### 2.1. Strain Construction and Growth Conditions

*B. anthracis* A16R strains were routinely cultured in LB medium. Sporulation was induced by nutrient exhaustion in Difco sporulation medium (DSM), with the start of sporulation (*T*_0_) defined as the end of exponential growth; *T*_n_ indicates *n* hours after *T*_0_ [5]. All strains and plasmids used in this study are listed in Table 1. All primers used for strain construction are listed in Appendix A.

The *spoVG* mutant strain was constructed as described previously [24] (Figure 1a). Briefly, vectors up-T, *spc*-T, and down-T and the acceptor vector pKMBKI were digested with *Bsa*I and then ligated. Subsequently, pKMUSD was introduced into strain A16R for homologous recombination, followed by introduction of plasmid pSS4332 and *spoVG* deletion screening.

A plasmid containing an amylase promoter, pBE2A, was used to construct the RΔ*spoVG* complementation strain; replication of the promoter region of *spoVG* on multicopy plasmids inhibits sporulation [25]. A fragment containing the complete *spoVG* open reading frame (ORF) was PCR-amplified from *B. anthracis* A16R genomic DNA. This fragment was then inserted into vector pBE2A to yield pBE2A*spoVG*. *Escherichia*
*coli* JM110 cells that were transformed with pBE2A*spoVG* to demethylate the plasmid. Subsequently, Δ*spoVG*-competent cells were transformed with demethylated pBE2A*spoVG* using electroporation (500 U, 25 mF, 0.6 kV) [26], which yielded the strain RΔ*spoVG* (Figure 1b).

In a similar manner, a fragment containing the *spoIIB* promoter region and ORF was PCR-amplified from *B. subtilis* strain 168. This fragment was inserted into pBE2, to yield pBE2*spoIIB_Bs_*. This vector was used to transform *B. anthracis* Δ*spoVG*-competent cells using the method described above, to give strain pB*spoIIB_Bs_*/Δ*spoVG* (Figure 1b).

### 2.2. Assay of Spore Formation Rate

Sporulation efficiency was determined as described previously [21,27]. In this method, cells cultured in DSM at 37 °C for 24 or 120 h were heat-inactivated. The number of spores was measured by measuring heat-resistant (70 °C for 30 min) colony-forming units (CFU) on LB-agar plates, while viable cells were measured as total CFU on LB-agar plates. Spore % = (spores/mL)/(viable cells/mL) × 100%.

### 2.3. Analysis of Sporulation Using Microscopy

Strains were stained with malachite green (a dormant spore-specific stain) and safranin O (a vegetative cell-specific stain) and examined using optical microscopy. Cells cultured in DSM for 24 or 120 h were collected and washed once with ddH_2_O, then resuspended in a small volume of ddH_2_O. Ten microliters of cell suspension were evenly smeared onto a glass slide and allowed to air dry. The air-dried cells were covered with a piece of filter paper, and a sufficient amount of 5% malachite green was added to the filter paper, which was baked under an alcohol lamp for 5 min. During this period, malachite green was periodically added to the slide to ensure that the filter paper did not dry. The back of the slide was washed with water for 1 min. The slide was stained with safranin O for 1 min, then washed again. After drying, slides were observed using a 100× objective lens.

### 2.4. β-Galactosidase Activity Assay

*B. anthracis* strains encoding *lacZ* under the transcriptional control of the *spoIIE* promoter were cultured in DSM at 37 °C with shaking at 220 rpm. From *T_−_*_1_ to *T*_17_ (where *T*_0_ is the end of the exponential phase and *T*_n_ is *n* hours after *T*_0_), 2 mL aliquots of cells were collected every 2 h, centrifuged at 12,000× *g* at 4 °C (3–30 k, Sigma, St. Louis, MO, USA), washed with low-salt phosphate-buffered saline, and stored at −80 °C. β-Galactosidase activity was determined as previously described [28,29]. The averages from at least three independent assays are reported. LacZ activity was calculated in Miller units [30].

### 2.5. RNA Isolation and Reverse Transcription Real-Time Quantitative PCR (RT-qPCR)

Total RNA was isolated from *B. anthracis* strains at the indicated time points using TRIzol Up reagent (Trans, Beijing, China). RNA concentrations were measured using a NanoDrop2000 instrument (Thermo, Wilmington, DE, USA). Total RNA was treated with DNase (Thermo, Vilnius, Lithuania) to digest genomic DNA. RNA was reverse transcribed into cDNA using EasyQuick RT MasterMix (CWBIO, Beijing, China) according to the manufacturer’s instructions. qPCR was performed using SYBR Green real-time master mix (CWBIO, Beijing, China) according to the manufacturer’s instructions.

### 2.6. Ultrastructural Studies of Sporulation Using Transmission Electron Microscopy (TEM)

A 1 mL sample of bacterial cells cultured in DSM for 24 h was collected and centrifuged at 12,000× *g* for 3 min at 4 °C. The supernatant was discarded and the pellet was washed twice with ddH_2_O. The pellet was resuspended in ddH_2_O, centrifuged at 4000× *g* for 5 min, and the supernatant was discarded. Two hundred microliters of 2.5% glutaraldehyde were added to the edge of the microcentrifuge tube and slowly mixed with the bacterial cells, then the mixture was incubated at 4 °C overnight. Electron microscopy was conducted as described previously [11].

### 2.7. Evaluation of Heat Resistance on LB-Agar Medium

*B. anthracis* strains A16R, Δ*spoVG*, RΔ*spoVG*, and pB*spoIIB_Bs_*/Δ*spoVG* were cultured in DSM for 24, 72, and 120 h. The cultures were serially diluted 10-fold (10^−1^ through 10^−5^) as described previously [31], and 10 μL aliquots from each dilution were plated on LB-agar plates. Cultures were heated to 70 °C for 30 min to inactive vegetative cells, and spores were plated using the method described above. Plates were incubated at 37 °C overnight and then photographed.

### 2.8. Phylogenetic Analysis

A phylogenetic tree was constructed based on alignments of the SpoVG and SpoIIB amino acid sequences from 10 strains using data obtained from NCBI (the National Center for Biotechnology Information; https://www.ncbi.nlm.nih.gov/). Phylogenetic trees were constructed using the unweighted pair group method with arithmetic mean (UPGMA) [32]. Trees were drawn to scale, with branch lengths in the same units as those of the evolutionary distances used to infer the phylogenetic tree. Evolutionary distances were computed using the Poisson correction method. Phylogenetic analyses were conducted using Mega X software [33].

### 2.9. Confocal Laser-Scanning Microscopy

Confocal laser-scanning microscopy was performed as previously described [23]. Briefly, cells were stained with the membrane dye FM4-64 (final concentration 100 µM; Molecular Probes, Inc., Eugene, OR, USA). Cells collected at the designated time points (*T*_1_ and *T*_3_ indicate the early stationary phase during spore development, and *T*_17_ indicates the late stationary phase) were observed under a confocal laser-scanning microscope (Carl Zeiss, Germany). Each strain was viewed in at least three fields.

## 3. Results

### 3.1. Deletion of spoVG Results in a Spore Formation Defect in B. anthracis

Although the absence of *spoVG* causes little impairment in sporulation in the model organism *B. subtilis*, it caused aberrations in the cortex [7,16] and amplified the effects of a *spoIIB* mutation [5,15]. To investigate whether SpoVG influenced sporulation in *B. anthracis* A16R, a *spoVG* mutant (Δ*spoVG*) and a plasmid-based complementation strain (RΔ*spoVG*) were constructed. The sporulation efficiency of the Δ*spoVG* strain was assessed via its heat-resistance (70 °C for 30 min) after cells were cultured in DSM for 24 h. The Δ*spoVG* mutant had a spore formation defect, with no heat-resistant spore forms observed (Figure 2a). The complemented strain RΔ*spoVG* showed partly restored sporulation, although the sporulation efficiency of RΔ*spoVG* was lower than that of the wild-type strain A16R. 

To further verify this mutant phenotype, cultures of the three strains were collected at 24 h and stained with malachite green and safranin O, followed by optical microscopy examination. Green spores were observed in cultures of both the wild-type strain A16R and the complemented strain RΔ*spoVG*, but not in the Δ*spoVG* culture (Figure 2b). Additionally, cultures of these three strains collected after 24 h were examined using TEM. TEM images of A16R and the complemented strain RΔ*spoVG* revealed that both formed intact spores (Figure 2c). However, TEM images of the Δ*spoVG* strain showed only rod-shaped, vegetative cells with no spores. These results indicated that deletion of *spoVG* results in a spore formation defect in *B. anthracis*.

### 3.2. Deletion of spoVG Resulted in a Complete Blockage Prior to Asymmetric Division in B. anthracis

As shown in Figure 2, no spores were formed by the ∆*spoVG* strain. To identify the sporulation stage at which blockage occurred, the membranes of live cells collected at *T*_1_ (8 h), *T*_3_ (10 h) and *T*_17_ (24 h) were stained with membrane-impermeable FM4-64 dye. Cells were observed using confocal microscopy, with the red fluorescent signal indicating the bacterial cell membrane. As indicated by arrows (Figure 3a), the wild-type strain A16R and the complemented strain RΔ*spoVG* showed complete asymmetric septum formation (yellow arrow) and the engulfment process (white arrow) at *T*_1_ and *T*_3_, respectively. Bright-field imaging showed that both strains produced many mature spores (red arrows) at *T*_17_, although the number of RΔ*spoVG* spores was lower than the number of A16R spores. However, the Δ*spoVG* strain did not form an asymmetric septum and showed no morphological changes. Therefore, deletion of *spoVG* prevented formation of an asymmetric septum.

Asymmetric division is the first obvious morphological feature of sporulation. The A16R cells underwent normal asymmetric division by *T*_1_, while ∆*spoVG* cells did not. To understand the physiological consequences of *spoVG* deletion at the proteomic level, total protein was collected from A16R and ∆*spoVG* cells at *T*_1_ and iTRAQ-based proteomic analysis (iTRAQ: isobaric tag for relative and absolute quantitation) was performed (Appendix A, Appendix A). The expression level of the stage II sporulation protein SpoIIE was significantly decreased in the Δ*spoVG* mutant (0.23-fold downregulation) (Appendix A). SpoIIE not only plays an important role in the formation of an asymmetric septum in *B. subtilis*, but is required for spore formation in *B. anthracis* [34,35]. To determine whether the *spoIIE* gene was responsible for the *B. anthracis* Δ*spoVG* mutant phenotype, the transcription levels and promoter activities of *spoIIE* were assayed. The mRNA expression level of *spoIIE* was significantly downregulated (50.63 ± 1.82-fold) in the Δ*spoVG* strain as measured using RT-qPCR (Figure 3b). Additionally, a P*_spoIIE_–lacZ* fusion was constructed. Following transformation into the wild-type strain A16R and the mutant strain Δ*spoVG*, this construct was used to assess transcription and regulation of the *spoIIE* promoter. β-Galactosidase assays showed that the transcriptional activity of P*_spoIIE_* in the Δ*spoVG* strain was much lower than that in the wild-type (Figure 3c). These results suggest that SpoVG positively regulates expression and transcription of *spoIIE*.

### 3.3. SpoIIB is Poorly Conserved Between B. anthracis and B. subtilis

In this study, we found that, unlike in *B. subtilis*, SpoVG was required for sporulation in *B. anthracis*. SpoVG, a pleiotropic regulatory factor, has little effect on spore formation in the model organism *B. subtilis* because of functional redundancy of SpoVG and SpoIIB [5]. To investigate whether this phenotypic difference is related to SpoIIB, similarity searching of SpoIIB from *B. subtilis* (SpoIIB_Bs_) was performed against the NCBI and Kyoto Encyclopedia of Genes and Genomes (KEGG) databases. We identified protein BA4688 and tentatively assigned this as SpoIIB in *B. anthracis*; among all *B. anthracis* proteins, BA4688 had the highest similarity to SpoIIB_Bs_, but the sequence similarity was low (27.9%). We aligned the amino acid sequences of SpoIIB and SpoVG from *Bacillus spp*. The amino acid sequence of SpoVG was highly conserved between *B.*
*anthracis*, *B. cereus*, *B. subtilis*, *B. thuringiensis*, and *B. amyloliquefaciens* (Figure 4a). However, the sequences of SpoIIB proteins from the *B. cereus* group species [36] were distinct from those of *B. subtilis* and *B. amyloliquefaciens* (Figure 4a). I.e., the amino acid sequences of SpoIIB_Bs_ and BA4688 are quite different.

In the genome of *B. anthracis*, the tentative *spoIIB* ORF (BA4688) was located between *folC* and *maF*. However, an additional gene, *comC*, was located between *folC* and *maF* adjacent to *spoIIB* in the genome of *B. subtilis* (Figure 4b). In *B. anthracis*, the *comC* gene (BA1318, NCBI Reference Sequence: NC_003997.3, region 1265988 to 1266710) is located between BA1317 and BA1319. The amino acid sequence of the ComC proteins of *B. anthracis* and *B. subtilis* share only 38.91% sequence identity (92% sequence coverage) (Appendix A). These results revealed that the amino acid sequence of SpoIIB was poorly conserved between *B. subtilis* and *B. anthracis*, and that the location of the *spoIIB* gene differed in *B. anthracis* and *B. subtilis*.

### 3.4. SpoIIB_Bs_ Could Not Restore the ΔspoVG Strain to Mature Resistant Spores

A single *spoVG* mutation had almost no effect on sporulation in *B. subtilis*; this phenotype was very different from that of the Δ*spoVG* mutant in *B. anthracis*. In *B. subtilis*, the Δ*spoVG*/Δ*spoIIB* double mutation prevented sporulation at the engulfment stage [20]. *spoIIB_Bs_* (the *spoIIB* gene of *B. subtilis*) was transferred into the *B. anthracis* ∆*spoVG* strain derived from strain A16R to explore whether SpoIIB_Bs_ could complement the sporulation deficiency of the *B. anthracis* ∆*spoVG* mutant. Spore formation efficiency was evaluated by culturing strains for 5 days in DSM (to ensure all spores developed to maturity). The pB*spoIIB_Bs_*/Δ*spoVG* strain, as well as the Δ*spoVG* mutant, showed no spore formation (Table 2). Additionally, the heat resistance of the pB*spoIIB_Bs_*/Δ*spoVG* strain was assayed at 24, 72, and 120 h, respectively. The pB*spoIIB_Bs_*/Δ*spoVG* strain did not form heat-resistant spores (Figure 5a). Spore staining showed that the pB*spoIIB_Bs_*/Δ*spoVG* strain did not form spores (green; Figure 5b). Thus, SpoIIB_Bs_ could not restore the ability of the *B. anthracis* Δ*spoVG* strain to form mature resistant spores.

### 3.5. SpoIIB_Bs_ Partially Restored Sporulation of ΔspoVG at the Engulfment Stage

Although the pB*spoIIB_Bs_*/Δ*spoVG* strain could not form mature resistant spores, a normal asymmetric septum (yellow arrow) was formed at *T*_1_. The pB*spoIIB_Bs_*/Δ*spoVG* strain subsequently underwent the prespore engulfment stage (white arrow) at *T*_3_ (Figure 6a). These results revealed that SpoIIB_Bs_ could partially restore the ability of the *B. anthracis* Δ*spoVG* strain to reach the prespore engulfment stage of sporulation, although the numbers of cells undergoing asymmetric division and engulfment were low.

To confirm this result, RNA was extracted from the A16R (wild-type), Δ*spoVG*, RΔ*spoVG*, and pB*spoIIB_Bs_*/Δ*spoVG* strains at *T*_17_. Levels of *spoIIM*, *spoIIQ*, and *cotE* mRNA were quantitated using RT-qPCR. Levels of *spoIIM* mRNA were significantly downregulated in Δ*spoVG* cells compared with the wild-type strain A16R (Figure 6b–d). Levels of *spoIIM* mRNA in the pB*spoIIB_Bs_*/Δ*spoVG* strain were almost identical to those in strain A16R. However, levels of *spoIIQ* and *cotE* mRNA had no significant change in the pB*spoIIB_Bs_*/Δ*spoVG* strain compared with those in strain ∆*spoVG*. These results indicated that SpoIIB_Bs_ could partially restore sporulation of the *B. anthracis* Δ*spoVG* strain at the engulfment stage.

## 4. Discussion

In this study, SpoVG was found to play a critical role in spore formation in *B. anthracis*. Sporulation was abolished, with blockage occurring before asymmetric septum formation, in the absence of *spoVG* (Figure 2 and Figure 3a). To investigate the impact of *spoVG* deletion on regulation of sporulation, total protein was collected from A16R (wild-type) and Δ*spoVG* cells at *T*_1_. iTRAQ-based proteomic analysis detected many differentially expressed proteins relating to sporulation (Appendix A). SpoIIE, an integral membrane protein, participates in asymmetric septum formation in *B. subtilis* [37,38], and *spoIIE* mutants cannot form spores in *B. anthracis* [35]. Expression of SpoIIE was significantly downregulated in the Δ*spoVG* mutant compared with the wild-type strain, as determined using iTRAQ analysis, and RT-qPCR showed that mRNA levels of *spoIIE* in the Δ*spoVG* strain were significantly downregulated compared with those in the wild-type (Figure 3b). P*_spoIIE_* showed almost no activity in the Δ*spoVG* strain in a β-galactosidase activity assay (Figure 3c). These data indicate that SpoVG has a positive regulatory effect on the *spoIIE* gene in *B. anthracis*; SpoVG affects sporulation by positively regulating *spoIIE* (Figure 7). This finding may provide a foundation for the investigation of a potential interaction between SpoVG and SpoIIE in *B. subtilis*.

It is interesting that the phenotype of the Δ*spoVG* strain of *B. anthracis* was completely different from that of the Δ*spoVG B. subtilis* mutant, even though the *spoVG* gene is highly conserved between these species (Figure 4a). In *B. subtilis*, the absence of *spoVG* from sporulating cells causes aberrations in the cortex, and defects were observed at stage V of sporulation [16]. Deletion of *spoVG* had little effect on spore formation in *B. subtilis*. Only double-mutation of *spoIIB* and *spoVG* strongly blocked sporulation, at the engulfment stage [11,39] (Figure 7a). Protein BA4688 is the most similar protein encoded in the *B. anthracis* genome to SpoIIB_Bs_, but the similarity is low, and the location of the putative *spoIIB* gene differed from that in the *B. subtilis* genome (Figure 4). Thus, we suspect that the low sequence similarity of SpoIIB results in the *B. subtilis* and *B. anthracis spoVG* mutants having different phenotypes.

To test this hypothesis, the *spoIIB_Bs_* gene was transferred into the *B. anthracis* Δ*spoVG* strain, producing strain pB*spoIIB_Bs_*/Δ*spoVG*. No resistant spores of this strain formed (Figure 5). In the pB*spoIIB_Bs_*/Δ*spoVG* strain, mRNA expression of the *spoIIM* gene was similar to that in the wild-type strain A16R (Figure 6b), which indicates that transcription of *spoIIM* was not impaired in the pBs*poIIB_Bs_*/Δ*spoVG* strain. However, the pB*spoIIB_Bs_*/Δ*spoVG* mutant showed low-level transcription of *spoIIQ* and *cotE* (Figure 6c,d). SpoIIM is required during the early stage of engulfment, and *spoIIM* mutants showed blocked sporulation prior to the completion of engulfment in *B. subtilis* [40]. Sporulation is blocked at the late stage of engulfment in *spoIIQ* mutants of *B. subtilis* [11]. CotE plays an essential role in assembly of both the coat and exosporium in *B. anthracis* and *B. cereus* spores [13,21,41,42]. Thus, SpoIIB_Bs_ may be able to stimulate sporulation of the *B. anthracis* pB*spoIIB_Bs_*/Δ*spoVG* strain to the engulfment stage (Figure 7b). This result provides evidence supporting our hypothesis—the poor sequence conservation of SpoIIB between *B. anthracis* and *B. subtilis* may be related to functional differences in SpoVG. The phenotypic differences of *spoVG* mutants of *B. anthracis* and *B. subtilis* may be one of the factors underlying *B. anthracis* pathogenicity. These data may help us further understand sporulation and provide new perspectives on *B. anthracis* pathogenesis.

Additionally, the *B. anthracis spoVG* mutant and the wild-type A16R strain showed similar growth rates during the exponential phase. However, cell densities of the Δ*spoVG* strain during the stationary phase were much lower than those of the wild-type (Appendix A), consistent with the results of heat-resistance assays (Figure 5a). Thus, we speculate that SpoVG might influence cell lysis rather than growth rate.

## 5. Conclusions

SpoVG was found to play an indispensable role in spore formation in *B. anthracis*. Our study serves to deepen understanding of the function of SpoVG in *B. anthracis*, and of the regulatory relationships of SpoVG (Figure 7). As illustrated in Figure 7b, SpoVG is involved in spore formation by modulating the expression of SpoIIE. Because of the low sequence similarity between *B. subtilis* SpoIIB (SpoIIB_Bs_) and BA4688 (the putative *spoIIB* of *B. anthracis*), we believe that SpoVG alone plays a crucial role in spore formation in *B. anthracis*, which is supported by our finding that sporulation was blocked before asymmetric division in the Δ*spoVG* strain derived from *B. anthracis* A16R. Our findings on the function of SpoVG in this study not only explain the functional mechanism of SpoVG in *B. anthracis*, but also increase our understanding of the phenotypic differences between *B. anthracis* and *B. subtilis*.

## Figures and Tables

**Figure 1 microorganisms-08-00548-f001:**
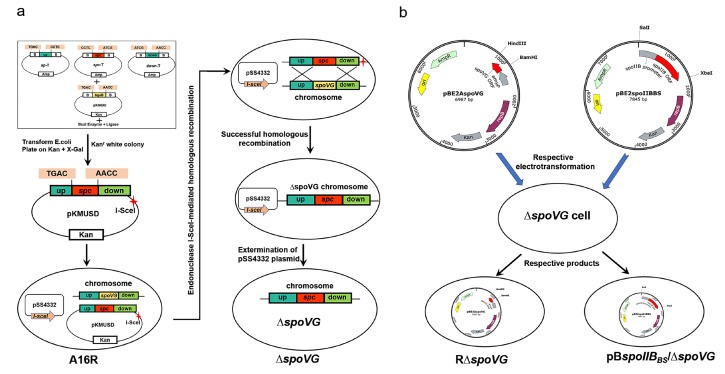
Schematic of construction of the *B. anthracis* Δ*spoVG* mutant and complemented strains. (**a**) Construction of the Δ*spoVG* strain. The construction method has been described in detail previously [24]. Briefly, the recombinant allelic exchange vector pKMUSD containing an *I-Sce*I site was constructed. Then, pKMUSD was introduced into *B. anthracis* strain A16R, followed by introduction of pSS4332. Expression of the endonuclease I-SceI from pSS4332 promotes homologous recombination. When homologous recombination is completed, pSS4332 is driven out, producing a ∆*spoVG* mutant. (**b**) The complementation plasmids pBE2A*spoVG* and pBE2*spoII**B_Bs_* were constructed as described in Materials and Methods. These plasmids were introduced into *B. anthracis* A16RΔ*spoVG* competent cells, to yield strains RΔ*spoVG* and pB*spoIIB_Bs_*/Δ*spoVG*, respectively.

**Figure 2 microorganisms-08-00548-f002:**
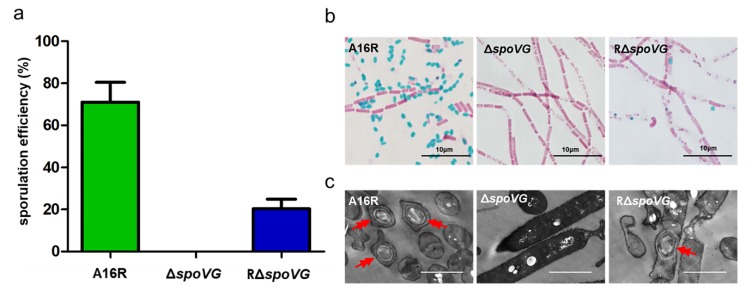
Role of SpoVG in sporulation. (**a**) Sporulation efficiency of *B. anthracis* strains A16R (wild-type), Δ*spoVG* and RΔ*spoVG* (complemented strain) cultured in DSM for 24 h. Values shown are means ± standard deviations (SDs) of triplicate experiments. (**b**) A16R, Δ*spoVG* and RΔ*spoVG* strain cultures (24 h) were stained with malachite green and safranin O. Spores and vegetative cells are stained green and red, respectively. Scale bar, 10 μm. (**c**) Ultrastructural observations of the three strains using transmission electron microscopy. Red double arrows indicate spores. Scale bar, 2 μm.

**Figure 3 microorganisms-08-00548-f003:**
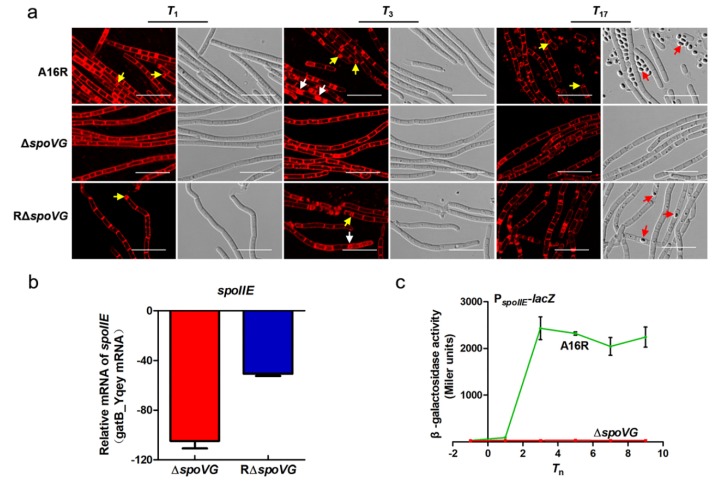
Defects in the formation of asymmetric septum in the *B. anthracis* Δ*spoVG* strain are associated with low transcriptional levels of *spoIIE*. (**a**) Confocal laser-scanning micrograph (scale bar, 10 µm) of the A16R (wild-type), Δ*spoVG*, and RΔ*spoVG* strains at *T*_1_, *T*_3_, and *T*_17_ (37 °C), *T*_n_ being *n* hours after *T*_0_ (the end of the exponential growth phase). The cell membrane is visible as red fluorescence. Yellow, white and red arrows indicate asymmetric septum, engulfed cells (prespores), and mature spores, respectively. (**b**) Relative mRNA expression of *spoIIE* determined in the Δ*spoVG* and RΔ*spoVG* strains compared with A16R (wild-type) cells at *T*_1_ using RT-qPCR. Values represent means ± SDs of triplicate experiments. (**c**) Transcription of P*_spoIIE_*–*lacZ* in A16R (wild-type) cells (green line) and Δ*spoVG*-mutant cells (red line) grown in DSM. Values represent the means of at least three independent replicates; error bars represent SDs.

**Figure 4 microorganisms-08-00548-f004:**
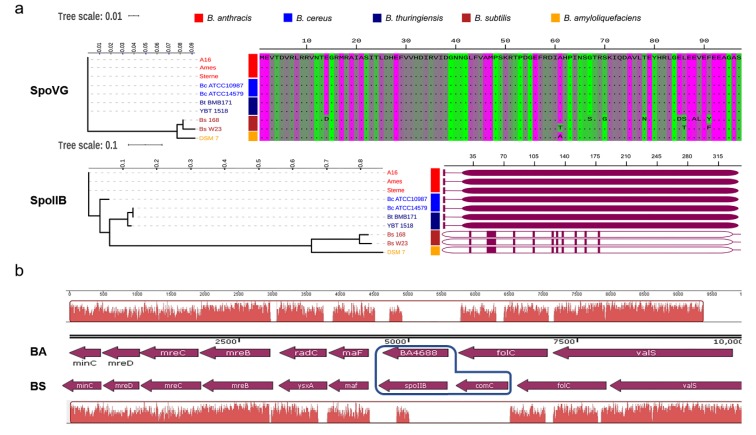
Amino acid sequence similarity of SpoVG and SpoIIB from species in the *B. cereus* group compared with those in *B. subtilis*. (**a**) Phylogenetic trees based on amino acid sequence alignment of SpoVG and SpoIIB, respectively. The unweighted pair group method with mean averages (UPGMA) tree was based on alignment of 10 amino acid sequences of SpoVG and SpoIIB proteins from strains belonging to the *B. cereus* group (seven sequences, including sequences from *B. anthracis*), *B. subtilis* (two sequences), and *B. amyloliquefaciens* (one sequence) available in the NCBI database (https://www.ncbi.nlm.nih.gov/). Multiple sequence alignment was conducted using ClustalX, and the tree was generated using Mega X software. The schematic shows regions of similarity rather than the complete sequence because of the length of the SpoIIB protein. SpoVG is highly conserved between *B. anthracis* and *B. subtilis*, while BA4688 (from *B. anthracis*) and SpoIIB_Bs_ (from *B. subtilis*) share a low amino acid sequence similarity. (**b**) Comparison of the *spoIIB* locus across *Bacillus* species. Arrows indicate the orientations of open reading frames. Original genome annotations are listed, and the names of organisms are abbreviated to: BA, *B. anthracis* Ames, and BS, *B. subtilis* 168. BA4688 represents the putative protein in *B. anthracis* that has the highest similarity to *B. subtilis* SpoIIB, but it still shares only a small region of similarity with the latter protein. There is some difference in the genetic location of BA4688 and *spoIIB_Bs_* between *B. anthracis* and *B. subtilis*.

**Figure 5 microorganisms-08-00548-f005:**
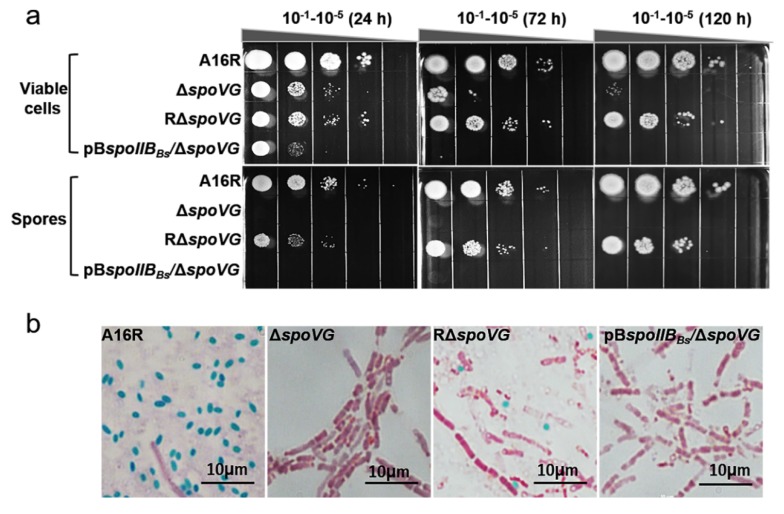
Sporulation of the *B. anthracis* pB*spoIIB_Bs_*/Δ*spoVG* strain. (**a**) Strains were cultured in DSM for 24, 72, and 120 h. The cultures were serially diluted and 10 μL aliquots of each dilution (10^−1^ through 10^−5^) were plated on LB-agar. After heat inactivation, spores were diluted and plated in the same manner. Images of plates after overnight incubation at 37 °C are shown. The Δ*spoVG* and pB*spoIIB_Bs_*/Δ*spoVG* strains did not form heat-resistant spores. (**b**) Cultures of four strains were stained with malachite green and safranin O at 120 h. Spores and vegetative cells were stained green and red, respectively. Scale bar, 10 μm.

**Figure 6 microorganisms-08-00548-f006:**
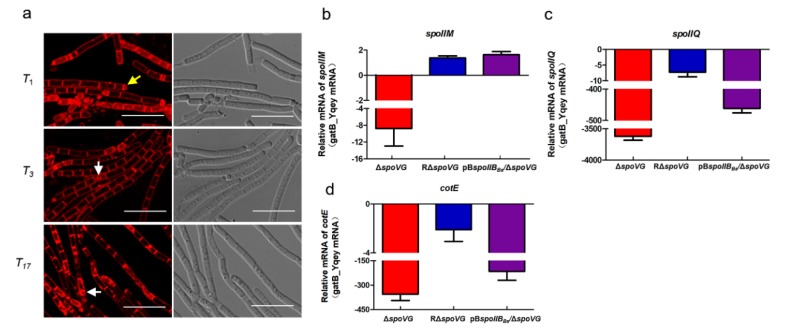
Morphological and molecular characteristics of the *B. anthracis* pB*spoIIB_Bs_*/Δ*spoVG* strain during sporulation. (**a**) Confocal laser-scanning micrographs (scale bar, 10 µm) of strain pB*spoIIB_Bs_*/Δ*spoVG* at *T*_1_, *T*_3_, and *T*_17_. The cell membrane is visible as red fluorescence. Yellow and white arrows indicate the asymmetric septum and engulfed cells (prespores), respectively. (**b****–d**) Relative mRNA expression of genes *spoIIM*, *spoIIQ*, and *cotE* determined in the Δ*spoVG*, RΔ*spoVG*, and pB*spoIIB_Bs_*/Δ*spoVG* strains compared with strain A16R (wild-type) at *T*_17_ using RT-qPCR. Values represent means ± SDs of at least two experiments.

**Figure 7 microorganisms-08-00548-f007:**
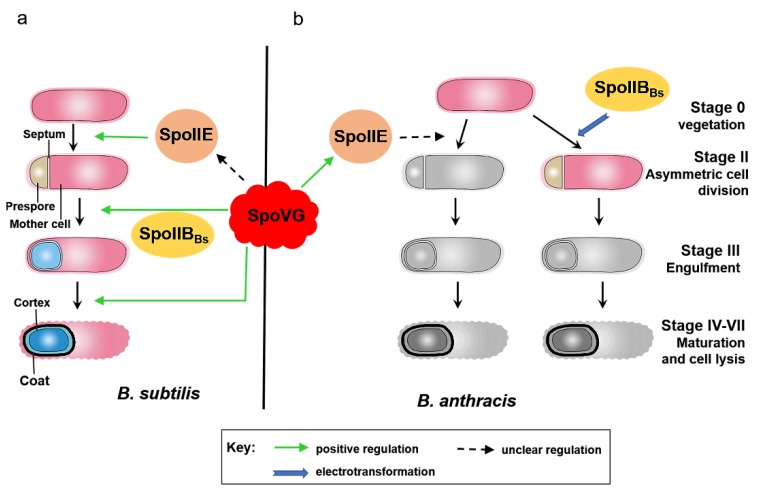
Schematic representation of regulatory pathways and their effectors in *B. anthracis* and *B. subtilis* based on the results of this study. (**a**) In *B. subtilis*, SpoVG is involved in spore formation at multiple stages (asymmetric division, engulfment, and cortex formation). Combined mutation of *spoIIB* and *spoVG* prevents spore formation at the engulfment stage. (**b**) In *B. anthracis*, the Δ*spoVG* mutant shows no asymmetric septum formation. SpoVG positively modulates spore formation through SpoIIE. *B. subtilis* SpoIIB (SpoIIB_Bs_) partly restored spore formation in the *B. anthracis* Δ*spoVG* strain at the engulfment stage of sporulation.

**Table 1 microorganisms-08-00548-t001:** Strains and plasmids used in this study.

Plasmid or Strain	Genotype or Description	Source
Plasmid		
pBE2A	Shuttle vector containing amylase promoter, Kan^r^ in *B. anthracis* and Amp^r^ in *Escherichia coli*	Our lab
pBE2	Shuttle vector, Kan^r^ in *B. anthracis* and Amp^r^ in *E. coli*	Our lab [21]
pBE2A*spoVG*	pBE2A carrying *spoVG* complete ORF, s*poVG* complementation plasmid, Amp^r^ in *E. coli*, Kan^r^ in *B. anthracis*	This study
pBE2*spoIIB_Bs_*	pBE2 carrying *spoIIB_Bs_*, *spoIIB* complementation plasmid, Amp^r^ in *E. coli*, Kan^r^ in *B. anthracis*	This study
pHT304	Shuttle vectors, Erm^r^, Amp^r^	Agaisse and Lereclus [22]
pHT304-*lacZ*	Promoterless *lacZ* vector, Erm^r^, Amp^r^, 9.7 kb	Fuping Song [23]
pHT304-P*_spoIIE_*	pHT304-*lacZ* carrying P*_spoIIE_*, Amp^r^ in *E. coli*, Erm^r^ in *B. anthracis*	This study
***E. coli***		
DH5α	F2, Q80d/lacZDM15, D(lacZYA-*argF*)U169, *deoR*, *recA*1, *endA*1, *hsdR*17(rk 2,mk + ), *phoA*, *supE*44l2, thi-1, gyrA96, relA1	Transgen, Beijing, China
JM110	*rpsL*(StrR), thr, leu, *endA*, thi-1, lacy, *galK*, *galT*, ara, *tonA*, tsx, dam-, dcm-, *supE*44(lac-*proAB*), F- [*traD*36, *proAB*, lacIqlacZΔM15]	Transgen, Beijing, China
***B. anthracis* strain**		
A16R	Human vaccine strain in China; derived from A16; pXO1^+^, pXO2^−^	Our lab [19]
Δ*spoVG*	A16R *spoVG* mutant, A16RΔ*spoVG*: *spc*	This study
RΔ*spoVG*	Δ*spoVG* genetic complementation strain containing pBE2A*spoVG* plasmid; Kan^r^	This study
pB*spoIIB_Bs_*/Δ*spoVG*	Δ*spoVG* genetic complementation strain containing pBE2*spoIIB_Bs_* plasmid; Kan^r^	This study
pHT304-P*_spoIIE_*/A16R	A16R strain containing plasmid pHT304-P*_spoIIE_*, Erm^r^ in *B. anthracis*	This study
pHT304-P*_spoIIE_/*Δ*spoVG*	Δ*spoVG* mutant strain containing plasmid pHT304-P*_spoIIE_*, Erm^r^ in *B. anthracis*	This study

**Table 2 microorganisms-08-00548-t002:** Effect of SpoIIB_Bs_ complementation on sporulation of the *B. anthracis* Δ*spoVG* strain.

Strain	Viable Cells ^a^ (CFU mL^−1^)	Spores ^a^ (CFU mL^−1^)	Spores/Viable Cells ×100(%)
A16R	1.34 × 10^7^	1.22 × 10^7^	91.04
Δ*spoVG*	2.39 × 10^4^	0	0
RΔ*spoVG*	2.73 × 10^5^	2.32 × 10^5^	84.98
pB*spoIIB_Bs_*/Δ*spoVG*	4.67 × 10^4^	0	0

^a^ The values in each column represent the average of three independent experiments.

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
