# Peer review of "SpoVG Is Necessary for Sporulation in Bacillus anthracis"

_microorganisms, 2020, doi:10.3390/microorganisms8040548_

Round 1

Reviewer 1 Report

This is a very well written manuscript in which the authors reveal the role of SpoVG in Bacillus anthracis sporulation. Sporulation in the SpoVG deletion mutant resulted in blocked sporulation prior to asymmetric septum formation. Importantly, this works reveals that the function of SpoVG in B. anthracis differs from that of B. subtilis; in the latter organism, the absence of spoVG caused no significant changes in sporulation efficiency because of the redundancy of spoVG and spoIIB. The material and methods are explained in detail.

Comment --> The English needs to be revised paying special attention to punctuation and the use of italics. Also, it would benefit from editing for clarity. Sometimes is difficult to follow the strain that the authors are talking about, the main conclusion of each section or why they do the next experiment. Trying to work a bit more on the flow of the story will significantly improve the manuscript.

Comment --> It appears that the complemented strain has greater viability and less ability to form spores. I would like the authors to suggest some hypotheses that can explain these results.

Comment --> This manuscript will be highly benefited from RNA studies of the wild-type strains and the knockout mutants (as well as the complemented strains) to better understand the results obtained. There is no need to do this in all strains used in this study but at least on the wild-type and knock out (qRT will be enough for the complemented strains). As it stands now the manuscript is highly interesting but raises many questions that a simple transcriptomic analysis would aid to understand with the purpose of generating hypotheses to follow up within the next paper.

Author Response

Please see the attachment 'Certificate of Editing'

Point 1: The English needs to be revised paying special attention to punctuation and the use of italics. Also, it would benefit from editing for clarity. Sometimes is difficult to follow the strain that the authors are talking about, the main conclusion of each section or why they do the next experiment. Trying to work a bit more on the flow of the story will significantly improve the manuscript.

Response 1:

Thank you for your patient suggestion. I am so sorry that my bit poor English and carelessness caused your difficult to understand. We have again edited the English text of the whole manuscript by the native English speaker from Liwen Bianji service. We have revised the punctuation and italics in this version. The description of the workflow has been added in the revised manuscript. And the main conclusion of each section has been revised careful in terms of word and logic. According to your suggestion, the revised manuscript will be easy to understand and improve.

Point 2: It appears that the complemented strain has greater viability and less ability to form spores. I would like the authors to suggest some hypotheses that can explain these results.

Response 2:

Thank you for your comment. The complemented strain was constructed by introducing pBE2AspoVG into ΔspoVG mutant. The efficiency of complement using plasmid exogenous expression is lower than in situ complement. This is also the case in other papers[1], and the complement strains using the multicopy plasmid can only partially restore the phenotype. In fact, the sporulation efficiency of the complemented strain is a little lower than those of wild-type in Table 2. The viable cells of the complemented strain is 2.73×105 CFU/mL in 120 h, to a lesser extent than those of the wild-type strains(1.34×107 CFU/mL). The possible explanations for these lower levels, though doubtful, are the polar effects due to the knockout disruption and the replication burden of the multicopy plasmid.

Point 3: This manuscript will be highly benefited from RNA studies of the wild-type strains and the knockout mutants (as well as the complemented strains) to better understand the results obtained. There is no need to do this in all strains used in this study but at least on the wild-type and knock out (qRT will be enough for the complemented strains). As it stands now the manuscript is highly interesting but raises many questions that a simple transcriptomic analysis would aid to understand with the purpose of generating hypotheses to follow up within the next paper.

Response 3:

Thank you for your constructive suggestions. We have done research on RNA, but unfortunately, the interference caused by 16s RNA in bacteria has led to the unreasonable results. As an alternative to it, the results of mass spectrometry are utilized in the manuscript, and the results of proteomics explain the difference between the wild-type strains and the knockout mutants to a certain extent. The next plan is to optimize the experiment conditions of the transcriptome to avoid 16s RNA interference, obtaining more interesting and valuable results.

  1. Bi, Changhao, Shawn W. Jones, Daniel R. Hess, Bryan P. Tracy, and Eleftherios T. Papoutsakis. "Spoiie Is Necessary for Asymmetric Division, Sporulation, and Expression of Sigmaf, Sigmae, and Sigmag but Does Not Control Solvent Production in Clostridium Acetobutylicum Atcc 824." Journal of Bacteriology 193, no. 19 (2011): 5130-37.

Reviewer 2 Report

Journal: Microorganisms

Manuscript ID: 749627

Title: SpoVG is Necessary for Sporulation in Bacillus anthracis

In this manuscript, a study on the role of SpoVG in sporulation of Bacillus anthracis was carried out. SpoVG is a pleiotropic regulatory factor, in Bacillus anthracis sporulation was impeded in ΔspoVG mutant.

Findings evidenced that, unlike Baillus subtilis, SpoVG appears to be required for sporulation in Bacillus anthracis, which provides further insights into the sporulation mechanisms of this organism. The study deals with a process that is still poorly understood, that of B. anthracis sporulation, which has hindered progress towards understanding its impacts on physiology and pathology.

In this study, Authors investigated the function of the spoVG gene in B. anthracis vaccine strain A16R (pXO1+, pXO2-) by constructing a ΔspoVG mutant. We found that in the ΔspoVG mutant, sporulation was blocked prior to asymmetric septum formation. Thus, the function of SpoVG in B. anthracis differs to that of B. subtilis; in the latter, the absence of spoVG caused no significant changes in sporulation efficiency because of the redundancy of spoVG and spoIIB.

The manuscript reports important results that can have an impact in the contexts of modalities of pathogenicity of B. anthracis, and iso f importance for the scientific community. Rigorous set of investigations evidenced the involvment of SpoVG in B. anthracis sporulation at level of the asimmetric septum.

Revision

Lines 158 and 159: ‘B. subtilis’ change to Italic style;

Figure 1: It is ifficult to comprehend the reported schema in Fig. 1. Please provide a clearer image;

Line 160: ‘B. anthracis’ , ‘ΔspoVG’ change to Italic style;

Line 162: ‘The sporulation efficiencies of ΔspoVG strains were assessed via their heat-resistance (70°C for 30 min) after cells were cultured in DSM for 24 h.’ In the cited reference (Grossman and Losick, 1988) and usually, it is used (80°C for 15 min). Of course it is correct, a depletion of temperature and an increase of time. Anyhow, why did you choose this temperature and this time?

Line 169: ‘ΔspoVG’ change to Italic style;

Lines from 173 to 180: ‘B. anthracis’ , ‘ΔspoVG’ change to Italic style;

Figure 4: The figure is not clear enough, please provide a better figure;

Figure 6: At line 291 it is reported ‘… the A16R wild-type strain (Fig. 6b–d).’, whereas in the legend of Fig. 6 it is mentioned it refers to Fig. 6 a) and b), only, without mentioning c) and d);

Lines from 381 to 486 (References section): the names of bacterial strains and more in general the binomial nomenclature must be observed, in Italic style and with the name of species in lowercase letter, as follows: ‘Bacillus Anthracis’ change to ‘Bacillus anthracis’ and in Italic style; ‘Bacillus Subtilis’ change to ‘Bacillus subtilis’ and in Italic style; ‘Bacillus Thuringiensis’ change to ‘Bacillus thuringiensis’ and in Italic style; ‘Bacillus Cereus’ change to ‘Bacillus cereus’ and in Italic style; ‘Clostridium Acetobutylicum’ change to ‘Clostridium acetobutylicum’ and in Italic style;

Supplementary material:

Line 95: ‘Paenibacillus Polymyxa’ _ ‘Bacillus Cereus’ change to ‘Paenibacillus polymyxa’ _ ‘Bacillus cereus’ and in Italic style.

Round 2

Reviewer 1 Report

The manuscript looks very good and the results are highly interesting. The revisions have significantly improved the manuscript.